# EPIC-Survival: End-to-end Part Inferred Clustering for Survival Analysis with Prognostic Stratification Boosting

**Hassan Muhammad**[*1,2]                         MUHAMMAH@MSKCC.ORG

**Chensu Xie**[*1,3]                                 XIEC@MSKCC.ORG

**Carlie S. Sigel**[1]                             SIGELC@MSKCC.ORG

**Michael Doukas**[4]                          M.DOUKAS@ERASMUSMC.NL

**Lindsay Alpert**[5]                    LINDSAY.ALPERT@UCHOSPITALS.EDU

**Amber Simpson**[6]                      AMBER.SIMPSON@QUEENSU.CA

**Thomas J. Fuchs**[7]                       THOMAS.FUCHS.AI@MSSM.EDU

[1] *Dept. of Pathology, Memorial Sloan Kettering Cancer Center, New York, USA*

[2] *Dept. of Physiology, Biophysics, and Systems Biology, Weill Cornell Medicine, New York, USA*

[3] *Tri-Institutional Training Program in Computational Biology & Medicine, New York, USA*

[4] *Dept. of Pathology, Erasmus Medical Center, Rotterdam, Netherlands*

[5] *Dept. of Anatomic Pathology, University of Chicago, Chicago, USA*

[6] *Dept. of Biomedical and Molecular Sciences , Queen's University, Ontario, Canada*

[7] *Dept. of Pathology, Hasso Plattner Institute for Digital Health at Mount Sinai, New York, USA*

## Abstract

Histopathology-based survival modelling has two major hurdles. Firstly, a well-performing survival model has minimal clinical application if it does not contribute to the stratification of a cancer patient cohort into different risk groups, preferably driven by histologic morphologies. In the clinical setting, individuals are not given specific prognostic predictions, but are rather predicted to lie within a risk group which has a general survival trend. Thus, It is imperative that a survival model produces well-stratified risk groups. Secondly, until now, survival modelling was done in a two-stage approach (encoding and aggregation). EPIC-Survival bridges encoding and aggregation into an end-to-end survival modelling approach, while introducing stratification boosting to encourage the model to not only optimize ranking, but also to discriminate between risk groups. In this study we show that EPIC-Survival performs better than other approaches in modelling intrahepatic cholangiocarcinoma (ICC), a historically difficult cancer to model. We found that stratification boosting further improves model performance and helps identify specific histologic differences, not commonly sought out in ICC.

**Keywords:** computational pathology, clustering, disease staging, survival analysis

## 1. Introduction

Cancer subtyping has shown to be uniquely powerful for survival analysis by many works (Delahunt et al., 2012). Because traditional methods used for discovering cancer subtypes are extremely labor intensive and subjective, successful stratification of common cancers, such as prostate, into effective subtypes has only been possible due to the existence of large datasets. However, working with rare cancers poses it's own set of challenges. Further,

---

[*] Contributed equally

histologic features are limited to the discretion of the manual observer's past experiences and subjectivity. EPIC-Survival offers a way to standardize cancer subtyping and discover new histologic features, as a unique deep learning-based survival model which overcomes two key barriers.

Firstly, even the best performing survival models are not useful unless they can provide stratified patient groups. It is difficult to computationally predict the specific outcome of an individual patient. It is more reasonable to predict the subgroup of a cancer population in which an individual patient falls into. Further, without a robust prognostic model which learns the population dynamics between histology and patient outcome or treatment prediction, survival models have minimal use. Thus, it is important that a survival model produces stratified groups, preferably driven by histology, rather than simply performing well at ranking patients by risk. Regardless, survival modelling based on whole slide image (WSI) histopathology is a difficult task which requires overcoming a second problem.

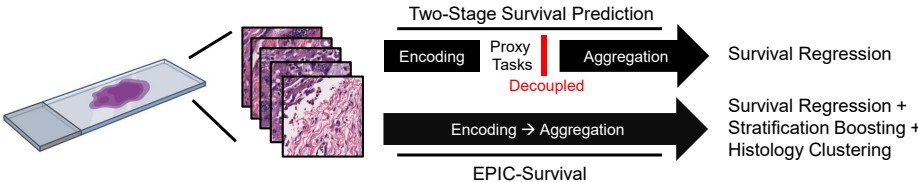

Figure 1: EPIC-Survival introduces end-to-end learning for prognostic prediction.

Because a single digitized WSI can span billions of pixels, it is impossible to directly use WSIs in full to train survival models, given current technological constraints. Thus, it is a common technique to sample tiles from WSIs, often in creative ways, and then aggregating them to represent their respective WSIs in the final step of training. We can simplify these stages as the tile *encoding* stage and the *aggregation* stages. While the aggregation stage of survival modelling has historically defaulted to the Cox-proportional Hazard regression model, recent advancements have made survival modelling more robust to complex data. We highlight some examples in the next section. Nevertheless, creative ways to extract features from WSIs and more advanced techniques to aggregate them still face the limits of operating in detached two-stage frameworks, in which the information at slide level, e.g. the given patient prognosis, is never taken into consideration while learning tile encoding by proxy tasks (cf. Figure 1). This creates a difficulty in being able to confidently identify specific and direct relationships between tissue morphology and patient prognosis, even though prognostic performance may be strong.

In this paper, we introduce a deep convolutional neural network which utilizes end-to-end training to directly produce survival risk scores for a given WSI without limitations on image size. Further, we contribute a new loss function called *stratification boosting* (SB) which further strengthens risk group separation and overall prognostic performance. Our introduction of SB not only improves overall performance, but also forces the model to identify risk groups. In contrast, other works attempt to find groups in the distribution of ranking after modelling a dataset. We claim that this model takes us one step closer to systematically mapping out the relationships between tissue morphology and patient death or cancer recurrence times. To challenge our method, we consider the difficult case of small dataset rare cancers.

## 1.1. Cholangiocarcinoma

cholangiocarcinoma (ICC), a cancer of the bile duct, has an incidence of approximately 1 in 160,000 in the United States (Saha et al., 2016). In general, the clinical standard for prognostic prediction and risk-based population stratification relies on simple metrics which are not based on histopathology. These methods have unreliable prognostic performances (Buettner et al., 2017), even when studied in relatively large cohorts (1000+ samples). Studies which have attempted to stratify ICC into different risk groups based on histopathology have been inconsistent and unsuccessful (Aishima et al., 2007; Nakajima et al., 1988; ?).

## 1.2. Related Works

Because survival analysis continues to operate in a two-stage approach as outlined above, advancements in survival analysis largely lie in the feature extraction front. (Muhammad et al., 2019). introduced a deep unsupervised clustering autoencoder which stratified a limited set of tiles randomly sampled from WSIs into groups based on visual features at high resolution. These clusters were then visualized and used as covariates to train simple univariate and multivariate CPH models. Similarly, in another study by Abbet et. al., self-supervised clustering was used to produce subtypes based on histologic features (Abbet et al., 2020). These were then visualized and used as covariates in survival models to measure significance of the clustered morphologies. (Zhu et al., 2017) takes the clustering approach one step further by modeling local clusters for a tile-level prediction before aggregating the results into slide-level survival predictions. These methods work to build visual dictionaries through clustering without having direct association to survival data. Slightly differently, (Yao et al., 2019) developed a method to build a visual dictionary through multiple instance learning. Though not completely unsupervised, even weak supervision can only operate with a decoupled survival regression. Other studies such as (Tabibu et al., 2019; Courtiol et al., 2019; Kather et al., 2019) have used even simpler approaches, producing models which learn to predict prognosis on tiles based on slide-level outcomes and then aggregate them into a slide-level predictions. These models, however, do utilize the DeepSurv (Katzman et al., 2016) function, a neural-network based survival learning loss robust to complex and non-linear data (discussed further in section 2.2). Unfortunately, the simplified feature extraction methods of the works listed do not allow the DeepSurv model to operate in its fullest potential—our method overcomes this barrier.

Recently, (Xie et al., 2020) bridged the gap of the two-stage problem in WSI classification tasks with the introduction of End-to-end Part Learning (EPL). EPL maps tiles of each WSI to $k$ feature groups defined as parts. The tile encoding and aggregation are learned together against slide label in an end-to-end manner. Although the authors suggested that EPL is theoretically applicable to survival regression, treatment recommendation, or other learnable WSI label predictions, the effort has been limited to testing the EPL framework with experiments benchmarking against classification datasets. In this study, we introduce EPIC-Survival to extend the EPL method to survival analysis by integrating the DeepSurv survival function, unencumbered by the limitations of two-stage training. Moreover, we contribute a new concept called stratification boosting, which acts as a critical loss term to the learning of distinct risk groups among the patient cohort.

## 2. Methods

### 2.1. Survival Modelling

Survival modelling is used to predict ranking of censored time-duration data. A sample is defined as censored when the end-point of its given time duration, or time-to-event, is not directly associated to the study. For example, in a dataset of time-to-death by cause of cancer, not all samples will have end-points associated with a cancer-related death. In some cases, an end-point may indicate a patient dropping out of the study or dying of other causes. Rather than filtering out censored samples and regressing only on uncensored time-to-events, Cox-propotional hazard (CPH) models are used to regress on a complete dataset and predict hazard, the instantaneous risk that the event of interest occurs. CPH as defined as:

$$\lambda(t) = \lambda_o e^{\beta_i v_i}, \tag{1}$$

where $\lambda(t)$ is the hazard function dependent on time $t$, $\lambda_o$ is a baseline hazard, and some covariate(s) $v_i$ are weighted by coefficient(s) $\beta_i$.

In 2016, DeepSurv (Katzman et al., 2016) made an advancement in survival modelling by using a neural network to regress survival data based on theoretical work proposed in 1995 (Faraggi and Simon, 1995). Their results showed better performance than the typical CPH model, especially on more complex data. In the case of a neural network-based survival function, $\beta_i$ is substituted for model parameters, $\theta$, i.e. $\beta_i v_i \rightarrow f_\theta(S)$, where $S$ represents the input slide image. Traditionally, a negative log partial likelihood (NLPL) is used to optimize the survival function. It is defined as:

$$NLPL(f_\theta(S), d, e) = - \sum_{i:E_i=1} (f_\theta(S_i) - log \sum_{j \in \Re(T_i)} e^{f_\theta(S_j)}), \tag{2}$$

where $f_\theta(S_i)$ is the output risk score for slide $i$, $d$ and $e$ are respective duration and event indicator, $f_\theta(S_j)$ is a risk score from ordered set $\Re(T_i) = i : T_i \geq t$ of patients still at risk of failure at time $t$, and $i : E_i = 1$ is the set of samples with an observed event (uncensored). The performance of a CPH or CPH-based model can be tested using a concordance index (CI) which compares the ranking of predicted risks to associated time-to-events. A CI of 0.5 indicates randomness and a CI of 1.0 indicates perfect prognostic predictions.

Further, the Kaplan-Meier (KM) method can be used to estimate a survival function, the probability of survival past time $t$, allowing for an illustrative way to see prognostic stratification between two or more groups. The survival function is defined as:

$$S(t) = \prod_{t_i < t} \frac{n_i - o_i}{n_n}, \tag{3}$$

where $o_i$ are the number of observed events at time $t$ and $n_i$ are the number of subjects at risk of death or recurrence prior to time $t$. The Log-Rank Test (LRT) is used to measure significance of separation between two survival functions modelled using KM. LRT is a special case of the chi-squared test used to test the null hypothesis that there is no difference between the $S(t)$ of two populations.

## 2.2. EPIC Survival

EPIC-Survival bridges the DeepSurv loss with the comprehensive framework of EPL. EPL models each WSI as $k$ groups of tiles with similar features, defined as parts, and backpropagates the loss against slide labels (time-to-event data) through the integrated encoding-aggregation graph, in which $k$ encoders ($\theta_e$) take in part representative tiles ($X$) and output part features ($z_i$) that are then concatenated and fed through a single fully connected aggregation layer ($\theta_a$). In each iteration, model weights were optimized and thus the centroid feature ($z_k = 1/N \sum_{n=1}^{N} \theta_e(x_{k,n})$) for each part was modified, then a tiles will be reassigned to parts and a different representative tile for each part will be selected for next iteration. Refer to (Xie et al., 2020) for mathematical inductions and training details. For EPIC-Survival, the last fully connected layer of the original EPL was replaced by a a series of fully connected layers and a single output node which functions as a risk score for a given input WSI. Similar to the traditional EPL, NLPL is combined with a clustering function based on minimizing distances between a sample embedding and its assigned centroid:

$$Loss = NLPL(f_\theta(S), d, e) + \lambda_c \sum_{i=1}^{N} ||z_i - c_i||^2, \tag{4}$$

where $z_i$ is the embedding of tiles sample from cluster $i$, $c_i$ is the part centroid from which $z_i$ is sampled, and $\lambda_c$ is a weighting parameter. Figure 2 visualizes this combined loss function, and slide-level and global clustering of visual morphology.

## 2.3. Stratification Boosting

While CPH and DeepSurv regressions serve to optimize the ranking of samples in relation to time-to-event data, they do not actively form risk groups within a dataset. In Mayr and Schmid's work on CI-based learning, they conclude that "specifically, prediction rules that are well calibrated do not necessarily have a high discriminatory power (and vice versa)" (Mayr and Schmid, 2014). One of the most important applications of survival analysis is cancer subtyping, an important tool used to help predict disease prognosis and direct therapy. Moreover, subtyping based on survival analysis creates a functional use for the survival model, especially if specific morphologies can be identified within each prognostic group. The DeepSurv loss, which only optimizes ranking, does not explicitly put a lower bound to the separation between the predicted risks. To further improve prognostic separation between high and low risk groups in the patient population, we extend the DeepSurv-EPL function with a stratification loss term. During training, predicted risks are numerically ordered and divided into two groups based on the median predicted risk. The mean is calculated for each group of predicted risks ($R_{high}$ and $R_{low}$) and the model is optimized to diverge the two values using Huber loss $smoothL_1(1/(1 + |R_{high} - R_{low}|), 0)$.

## 2.4. Dataset

WSIs of ICC cases were obtained from Memorial Sloan Kettering Cancer Center (MSKCC), Erasmus Medical Center-Rotterdam (EMC), and University of Chicago (UC) with approval from each respective Institutional Review Boards. In total, 265 patients with resected ICC without neoadjuvant chemotherapy were included in the analysis. Up-to-date retrospective

data for recurrence free survival after resection was also obtained. A subset of samples (n=157) from MSKCC were classified into their respective AJCC (Farges et al., 2011) TNM and P-Stage groups. 246 slides from MSKCC and EMC were used as training data, split into five folds for cross validation. 19 slides from UC were set aside as an external held-out test set. Using a web-based whole slide viewer developed by our group, areas of tumor were manually annotated in each WSI. Using a touchscreen tablet and desktop (Surface Pro 3, Surface Studio; Microsoft Inc.), a pathologist painted over regions of tumor to identify where tiles should be extracted for training. Tiles used in training were extracted from tumor-regions of tissue and sampled at 224x224px, 20x resolution.

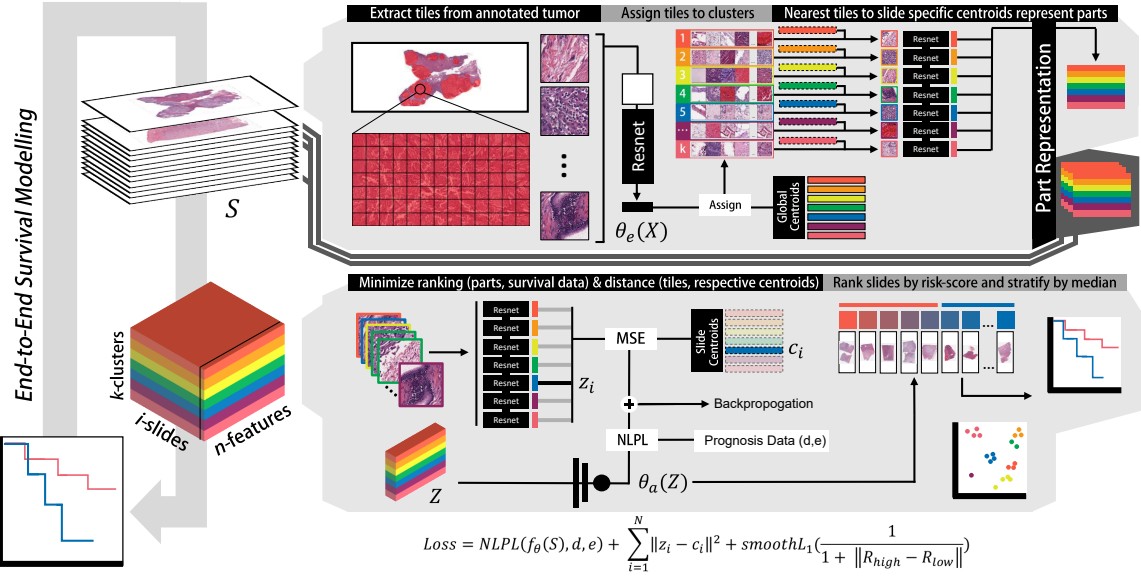

Figure 2: Diagram of the proposed EPIC-Survival approach for prognosis prediction. **Top:** Whole slide images are tiled into small patches which pass through an ImageNet ResNet-34 backbone, outputting a tile feature vector. Each vector is assigned to a histology feature group defined by global centroids. Next, local slide-level centroids are calculated and the nearest tiles to $k$ local centroids are used as part representations of the slide. This process is repeated for all slides. **Bottom:** Still within the same training epoch, parts of all slides are concatenated and trained with survival data, in conjunction with optimizing local clustering and overall risk group separation. Note: Global centroids are randomnly initialized before training and updated between epochs, based on the optimization of the ResNet-34 backbone.

## 2.5. Architecture and Experiments

An ImageNet ResNet-34 was used as the base feature extractor ($\theta_e$). A series of three wide fully connected layers (4096, 4096, 256) with dropout were implemented before the single risk output node. Model hyperparameters (number of clusters, waist size, part-batch size, learning rate, dropout rate, and top-k tiles respectively) were optimized using random grid search and CI as a performance metric at the end of each epoch. 16 clusters and a waist size of 16 produced the best performance. The same 5-fold cross validation was implemented and held throughout all experiments and models. Predicted risks of the validation sets from each fold were concatenated for a complete performance analysis using CI and LRT. Each model was subsequently trained using all training data, tested on the held-out test set, and evaluated using CI and LRT.

As a baseline, Deep Clustering Convolutional Autoencoder (Muhammad et al., 2019) was implemented. This model was chosen because, like EPIC-Survival, it uses clustering to define morphological features. However, these features are learned based on image reconstruction and then used as covariates in traditional CPH modelling, as a representation for the classic two-stage approach. Further, the subset of training data with AJCC staging, a clinical standard, was analyzed using a 4-fold cross validation and CPH.

## 3. Results

EPIC-Survival with and without SB performed similarly on the 5-fold cross validation producing CI of 0.671 and 0.674, respectively. On the held out test set, EPIC-survival with SB performed significantly better with a CI of 0.880, compared to a CI of 0.652 without SB. Unsupervised clustering with a traditional CPH regression yielded a CI of 0.583 on 5-fold cross validation and 0.614 on the test set. Table 1 summarizes these results.

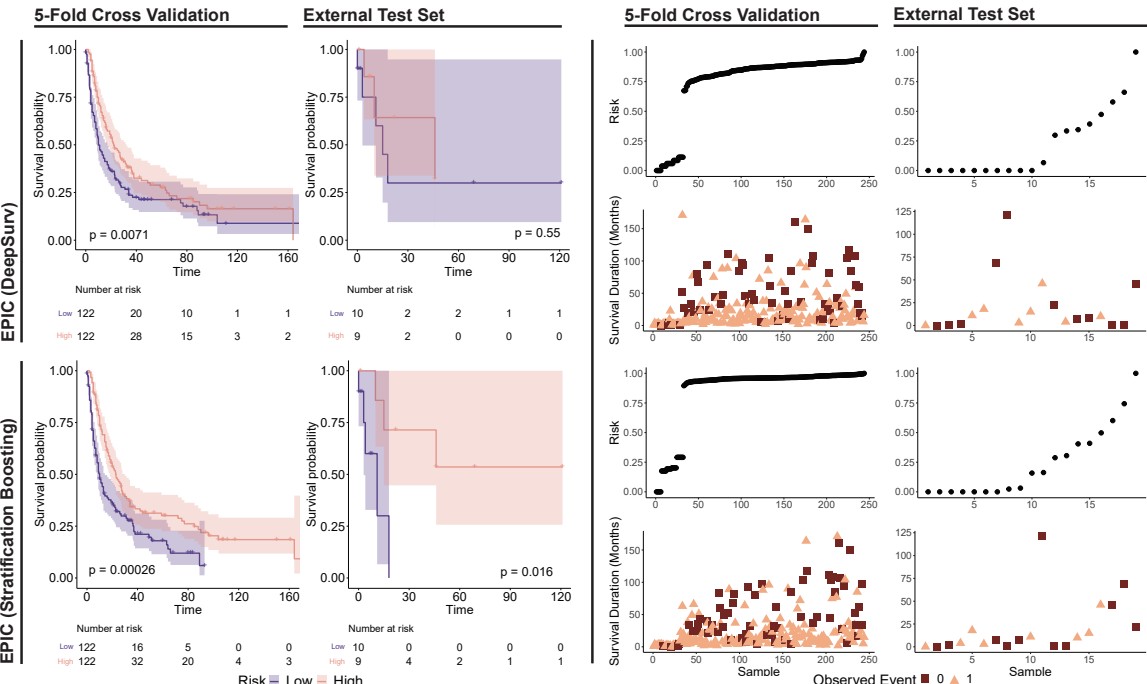

Figure 3: **Left:** EPIC-Survival without stratification successfully stratifies (LRT: p < 0.05) the patient population into high and low risks on 5-Fold Cross Validation but fails on the held out test set. Stratification boosting (SB) produces strong patient population separation on both 5-Fold Cross Validation and the External Test Set. **Right:** We visualize the distribution of time-to-events relative to predicted risk scores ordered from low to high. EPIC-Survival, in general, does well at predicting early recurrence. The inclusion of SB improves the correlation between predicted risk values and patient outcome.

AJCC staging using the TRN and P-stage protocols on the subset of ICC produced CIs of 0.576 and 0.638, respectively. While we recognize that a CI produced on a subset of data may produce biases from batch effects, these results are not different from the results of a study which tested multiple prognostic scores on a very large ICC cohort (n=1054) (Buettner et al., 2017).

In a KM analysis (Figure 3), EPIC-Survival with SB showed significant separation between high and low risk populations ($p < 0.05$). Epic-Survival without SB failed on the held out test set. Although stratification on the 5-fold cross validation is assumed significant, there remains a risk of crossing survival curves, breaking the assumption of proportional hazard rates.

To further analyze results, we visualize the distribution of predicted risks relative to the distribution of time-to-events (Figure 3). We found that EPIC-Survival with and without SB performs well at predicting early recurrence (<50 months). Correlation between predicted risks and time durations of the external test set using EPIC-Survival with SB is very strong, as further indicated by the strong CI of 0.880.

|  | Cross Validation | Test |
| --- | --- | --- |
| AJCC TNM | 0.576 (n=157) | - |
| AJCC P-Stage | 0.638 (n=157) | - |
| Muhammad et. al. | 0.583 (n=244) | 0.614 (n=19) |
| EPIC (DeepSurv) | 0.671 (n=244) | 0.652 (n=19) |
| EPIC (Stratfication Boosting) | **0.674** (n=244) | **0.880** (n=19) |
| AJCC TNM | - | 0.582 (n=1054) |
| Wang Nomogram | - | 0.607 (n=1054) |
| LCSGj | - | 0.562 (n=1054) |
| Okabayashi | - | 0.557 (n=1054) |
| Nathan Staging | - | 0.581 (n=1054) |
| Hyder Nomogram | - | 0.521 (n=1054) |

Table 1: EPIC-Survival with stratification boosting showed the best CI-based performance. For reference, performance of various clinical metrics on a very large ICC dataset (n=1054) are provided (Buettner et al., 2017).

In Appendix A, we visualize part representation (rows) in each slide (columns) from the test set. The slides are ordered by predicted risk scores. A gastrointestinal pathologist reviewed these and discovered some general trends indicating that tiles with a low predicted risk (earlier rate of recurrence) tended to have loose, desmoplastic stroma with haphazard, delicate collagen fibers, whereas high risk tiles (later recurrence) tended to have dense intratumoral stroma with thickened collagen fibers. The quality of nuclear chromatin was vesicular more commonly in the low risk tiles. The quality of the intratumoral stroma has never been a part of tumor grading or observed as a prognostic marker. Further, there is no grading scheme that involves assessment of nuclear features for ICC.

## 4. Conclusion

Our test results show a significantly higher CI than the cross validation experiments. We found that CI on smaller sets are often larger because correctly ranking a smaller set of data is easier. During hyperparameter optimization, this was also observed in the case of batch sizes. Smaller batch sizes produces better CIs—in other words, optimizing the ranking of smaller batches was easier than optimizing the ranking in larger batches.

EPIC-Survival has the capacity to identify specific risk factors in histology, though these morphologies would need further testing on a larger study. We hypothesise that altering the SB component of the loss function to push separation between >2 groups would further improve performance and has the potential to function as a general subtyping model.

Our contributions are threefold: (1) we introduce the first end-to-end survival model, overcoming the information decoupling-limitation of two-stage approaches; (2) we contribute a new loss term to strengthen the traditional hazard regression and encourage the learning of stratified risk groups; (3) we show the power of EPIC-Survival by applying it to the difficult test case of ICC, surpassing other metrics and providing insight into new histologic features which may unlock new discoveries in ICC subtyping.

## Acknowledgments

T.J.F. is co-founder and equity holder of Paige.AI. H.M. and T.J.F. have intellectual property interests relevant to the work that is the subject of this paper. MSK has financial interests in Paige.AI and intellectual property interests relevant to the work that is the subject of this paper.

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

## Appendix A. Predictive Histology

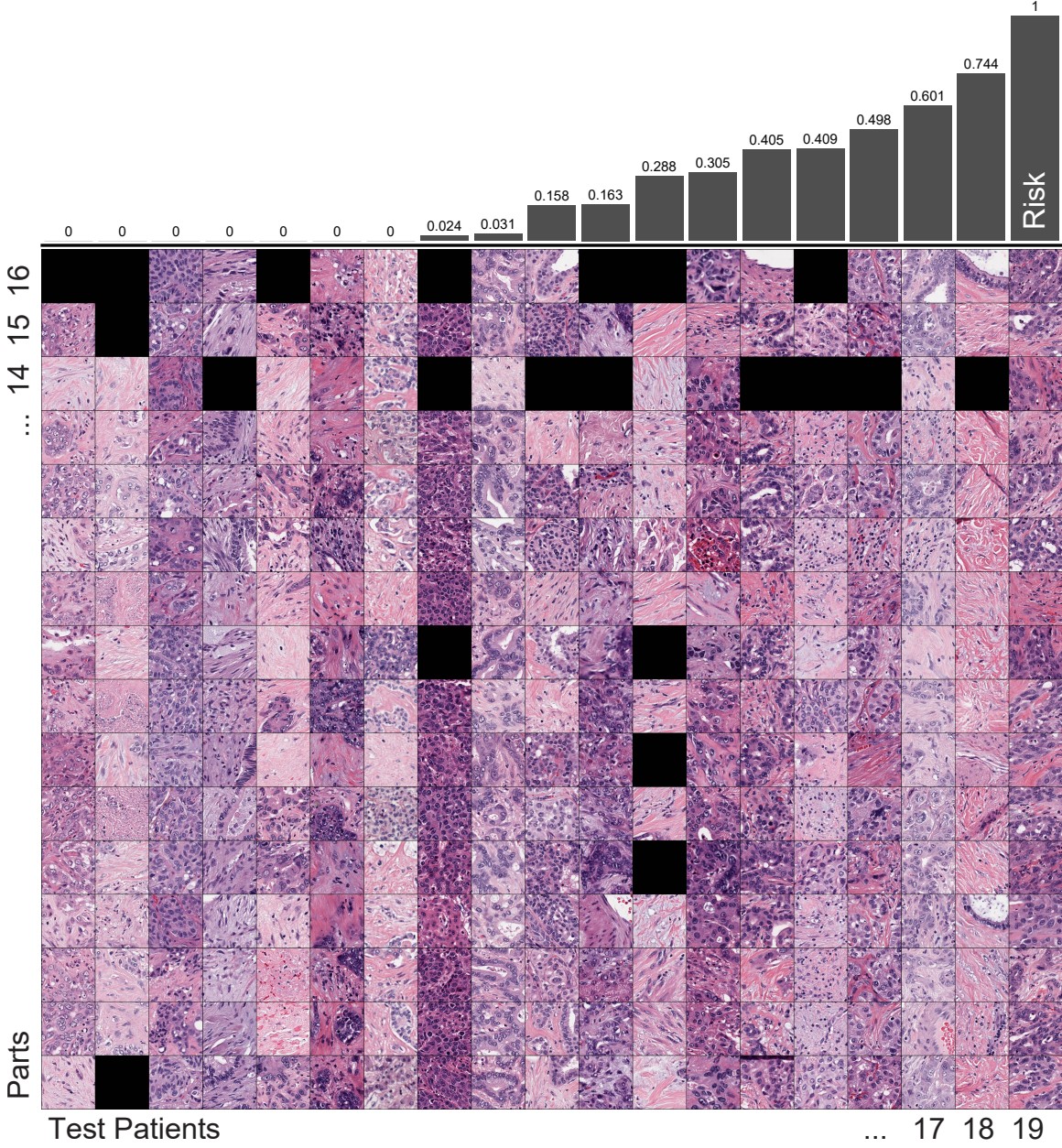

Figure 4: **Rows:** Slide parts, **Columns:** Patients with their predicted risk scores highlighted above. Black tiles indicate that there was no assigned tile to that part of a slide.

