# OpenReview forum: "EPIC-Survival: End-to-end Part Inferred Clustering for Survival Analysis, with Prognostic Stratification Boosting"
_MIDL.io/2021/Conference — MIDL 2021_

### Official Review · AnonReviewer1 · 2021-03-04

**Confidence:** 3
**Preliminary Rating:** 3
**Final Rating:** 3

**Summary:**

The paper proposes an end-to-end approach for histopathology-based survival modeling, augmented by risk group stratification. The proposed approach has been applied to rare cancers and evaluated on 265 patients. Empirical results show that the proposed approach, together with stratication boosting, achieves the best concordance index.

**Strengths:**

* The end-to-end approach for survival analysis is well-motivated and supported by empirical results on ICC data.
* The stratification boosting significantly improves the concordance index on the held-out set.
* The resulting tiles of different parts are visually meaningful and can help further understanding of ICC subtyping.

**Weaknesses:**

1. Does the same conclusion generalize to larger datasets such as (Buettner et al., 2017)? This is a concern due to the small size of the test set—only 19 slides from UC. The 19 slides might contain characteristics of cancer that are not representative of the population. The authors should explain why choosing UC as the test set, rather than MSKCC or EMC. An alternative approach is to use MSKCC for training and validation and evaluate on UC and EMC separately.

2. Why there is a large gap between cross-validation and test performance when comparing EPIC (DeepSurv) and EPIC (Stratcation Boosting)? This might be related to 1 in that the patient distribution in UC is different from MSKCC and EMC.

3. In Stratication Boosting, the choice of splitting the data based on median seems rather arbitrary. A uniform split might not respect the distribution of subtypes.

**Deanonymize Review:**

no

**Detailed Comments:**

Very minor comment:
In section 2.3, opening quotes should be typeset properly in latex with ticks.




**Final Rating Justification:**

I maintain my rating since no rebuttal is submitted.

**Justification Of The Preliminary Rating:**

The proposed approaches on end-to-end survival analysis and stratification boosting are novel and shown to work well on a small cancer dataset. Considering the small size of the test set and concerns in its potential representativeness of the population, I tend to vote for Weak Accept.

**Paper Type:**

both

**Questions To Address In The Rebuttal:**

Please address my comments in the Weaknesses section.

**Special Issue:**

no

---

> ### Author Response · Authors · 2021-03-17
> **response to AnonReviewer1**
>
> Thank you for the thorough review of our paper. Based on your feedback, we clarified some language and address your concerns in detail below:
>
> 1. Does model generalize to large dataset such as referenced dataset
>
> Unfortunately, it is currently impossible to test our model against a large cholangio dataset. While our dataset is smaller than desired, it is still the largest study ever conducted on cholangio digital histopathology. The larger dataset we compare to in the results is not based on digital histopathology, but rather metrics derived from radiology, genomics, and clinical variables. Even with 1000+ samples, these metrics underperform our small-dataset model.
>
> UC was picked as a test set to measure generalization on external data which has different scanning and staining protocols. The validation of the model is not limited only to the small test set. In our cross validation method, all model parameter optimization was done on one fold. Afterwards, all 5 folds were trained and their validation risk predictions were combined to produce one aggregate result.
>
> 2. Why is there a large difference in test set and cross validation?
>
> A smaller set is likely to behave differently from a larger set. This is indicated by the large confidence bounds in the KM curves. This explains the benefit of stratification boosting. In a small dataset, it is more likely that simple ranking will not produce a good split, i.e, a distribution with distinct groups. Its easier to have a bad split of patients for small datasets even if there is a good ranking
>
> 3. Splitting the data on median.
>
> This is a great criticism also shared by the second reviewer. We are copying our response here:
>
> Splitting the patients into two groups at inference is completely arbitrary. We chose to always split at the median so we don’t influence the appearance of the KM plots. Theoretically, a better way to split the population into two groups would be based on distribution, similar to the Otsu method. Even better, a model can be trained to learn/optimize this threshold. Instead, our model is trained to encourage at least 2 distinct groups within the distribution of predicted sample risks. This distribution of risk scores can be divided in any way useful to the scenario.
>
> We chose not to split into 4 groups (quartiles) at inference because then the test set would have too little samples in each group. The number of groups to split into (and the positions to split them) at inference is up to the user.
>
> Learning multiple groups requires learning the threshold at which the risk scores are split. We hope to do this in our next project. For now, SB encourages the risk distribution to better separate high and low scores.
>
> --
> We appreciate the minor remark on quotation typesetting.
>
> PS We are in communication with our institution. Once we get approval, we will post full code.

---

> > ### Comment · AnonReviewer1 · 2021-03-19
> > **Thanks for the response**
> >
> > I maintain my rating after reading the response.

---

> ### Author Response · Authors · 2021-03-19
> **dear reviewer Official Review of Paper49 by AnonReviewer1**
>
> We posted our rebuttal before the deadline but didn't realize that there was a "view" setting. We just changed the "view" to public (it was only viewable to program chairs and authors before). Apologies for that. Please consider reviewing our rebuttal.

---

### Official Review · AnonReviewer4 · 2021-03-08

**Confidence:** 5
**Preliminary Rating:** 3
**Final Rating:** 3

**Summary:**

This paper presents an approach to train CNNs end-to-end to produce survival risk scores from digital pathology whole-slide images (WSI).
The method is based on the combination of previous work from the same group (i.e., the EPL approach) and from the DeepSurv model.
Furthermore, a new loss term is introduced, namely "stratification boosting", which has the potential to identify risk groups, which extends the basic combination of EPL and DeepSurv.
The method is trained and validated on a small multi-centric dataset of 265 cases of (rare) intrahepatic cholangiocarcioma.
Results are reported both using cross-validation on 246 cases and on the held-out set of 19 cases.
The authors show that the proposed approach, i.e., EPL + DeepSurv + stratification boosting performs better than EPL + DeepSurv on the 19 test cases.

**Strengths:**

* As claimed by the authors, this is the first end-to-end model for the prediction of survival from WSI, which is an important contribution to the field of digital pathology and oncology in general.
* The method computes centroids and detects representative patches, a feature inherited by the EPL method, that can provide some type of explanation about what the model considers as relevant to compute the risk.
* The results on the external test set show some capabilities of this model to distinguish ICC cases with different prognoses.

**Weaknesses:**

* The authors limit the development and validation of this method to a small dataset of rare cancer. Although the authors state that this was done to challenge the method, this also results in testing the method on a very small external test set, which I consider as one of the main limitations of this paper. When training a method to predict survival, one could take fairly larger datasets from, for example, TCGA, and consider diseases with larger sample sizes, as for example breast or lung cancer. Using cross-validation, both results with and without SB have a significant p-value (although much smaller when SB is used), and the difference shown on the external test set is based on a very small set, which potentially has little statistical power.
* The method processes slides at 10X magnification, which is not the maximum level of details that can be achieved with standard digital pathology scanners. No explanations are given why this choice was made, and whether higher resolutions would have given different results.
* Figure 2 looks nice, but I find it a bit confusing and difficult to follow the actual workflow.
* Patches are only extracted from tumor regions, initially manually annotated, but it is not clear how this is going to work at test time when possibly no manual annotations of the tumor are available.
* In the introduction, the authors introduce the SB method as a way to further discriminate patients into multiple risk categories, criticizing existing methods for not addressing this problem. However, in the end, the SB method is not used to come up with multiple groups in this paper, and this possibility is only mentioned in the conclusion as something that could be possibly done in the future. I think that this is a pity that this has not been addressed in this paper, as it would have shown the full potential of the SB function.
* In the conclusion, the authors mention that this approach overcomes memory limits of two-stage approaches, but nothing is shown to support this claim

**Deanonymize Review:**

no

**Final Rating Justification:**

I would like to thank the authors for clarifying some of my doubts.
I still believe that the method presented in this paper should have been validated on a larger dataset, to show the potential value of the proposed stratification boosting method to identify multiple patient categories.

**Justification Of The Preliminary Rating:**

The paper introduces a method to address an important research question in histopathology, partly basing the method on previous work and partly introducing some novelty, namely the SB function.
However, the effectiveness of this method, in particular when the SB function is used, is only shown on a very small set of 19 patients.
Furthermore, the novel SB function is not used at its "full potential" to find multiple risk categories.

**Paper Type:**

both

**Special Issue:**

no

---

> ### Author Response · Authors · 2021-03-17
> **response to AnonReviewer4**
>
> Thank you for the thorough review of our paper. Based on your feedback, we clarified some language and address your concerns in detail below:
>
> 1. Concerns around model validation
>
> The validation of the model is not limited only to the small test set. In our cross validation method, all model parameter optimization was done on one fold. Afterwards, all 5 folds were trained and their validation risk predictions were combined to produce one aggregate result. While this dataset is smaller than desired, it is still the largest study ever conducted on cholangiocarcinoma digital histopathology. The larger dataset we compare to in the results is not based on digital histopathology.
>
> The test set, though small, allows us to make sure that the model can generalize on a dataset from a different institution with different staining and scanning protocols
>
> 2. Tile generation at 10X
>
> This was an unfortunate typo and we thank you for pointing the issue with valid criticism. We did in fact generate tiles from 20x magnification. This resolution was decided after consulting with a pathologist and learning that they analyze cholangiocarcinoma at 20x. This has been corrected in the PDF.
>
> 3. Clarity of Figure 2
>
> Based on some thorough criticism from the first reviewer, we realized that mathematical notation was not consistent across equations. This has been corrected and further, this notation is now included within figure 2 with slight adjustments in the workflow of the second panel. Refer to the updated PDF. We hope that both our mathematics and visualization are now made clearer.
>
> 4. Patch extraction from tumor-regions.
>
> This model requires annotation for testing. Our Annotation is not difficult or time consuming. We do not use pixel wise annotation, but rather, a general area of tumor is quickly circled by the pathologist. We do not believe this step to be a hindrance to clinical application.
>
> 5. Using stratification boosting to generate multiple groups, as opposed to 2 groups
>
> This is a great point to address. First of all, splitting the patients into two groups at inference is completely arbitrary. We chose to always split at the median so we don’t influence the appearance of the KM plots. Theoretically, a better way to split the population into two groups would be based on distribution, similar to the Otsu method. Even better, a model can be trained to learn/optimize this threshold. Instead, our model is trained to encourage at least 2 distinct groups within the distribution of predicted sample risks. Risk scores can be split into groups in any way useful to the scenario.
>
> We chose not to split into 4 groups (quartiles) at inference because then the test set would have too little samples in each group. The number of groups to split into (and the positions to split them) at inference is up to the user.
>
> Learning multiple groups requires learning the threshold at which the risk scores are split. We hope to do this in our next project. For now, SB encourages the risk distribution to better separate high and low scores.
>
> 5. Our claim to overcome memory limits
>
> We did not mean to make that claim. It was an unfortunate error in how we worded that sentence.  We have rephrased that claim to “we introduce the first end-to-end survival model, overcoming the information decoupling-limitation of two-stage approaches”
>
> PS We are in communication with our institution. Once we get approval, we will post full code.

---

> ### Author Response · Authors · 2021-03-19
> **view setting**
>
> We posted our rebuttal before the deadline (indicated on timestamp) but didn't realize that there was a "view" setting. We just changed the "view" to public (it was only viewable to program chairs and authors before). Apologies for that. Please consider reviewing our rebuttal.

---

### Official Review · AnonReviewer3 · 2021-03-09

**Confidence:** 4
**Preliminary Rating:** 2
**Final Rating:** 3

**Summary:**

 This paper focuses on discussing survival analysis based on whole slide images (WSIs). The authors propose an end-to-end survival model under the deep convolutional neural network framework. The authors introduce stratification boosting in the model. As shown in the experimental results, stratification boosting helps improve the test performance of the model.

**Strengths:**

The authors propose a novel end-to-end survival model by analyzing WSIs. The authors introduce stratification boosting in the model. Experimental results show that stratification boosting improves the Concordance Index (CI) in the test set.

**Weaknesses:**

The motivations of the method are not clear to me, and I am unable to understand some details about the method.

As described in Equation (4), it looks like the method involves clustering. I am not sure why clustering is necessary. It seems possible to design an end-to-end model if we encode each of the tile samples and then aggregate the encoding results, without clustering. In Equation (4), it is also unclear how the embedding $z_i$ is related to the corresponding tile. Is it assumed that each tile can be reconstructed from $z_i$? Since Equation (4) involves clustering, the cluster indicator should be updated during training. This is also not obvious in the equation.

In Equation (4), it is also unclear how the embeddings $z_i$ are aggregated into the negative log partial likelihood (NLPL). NLPL should be written as a function of $z_i$.

The motivation of introducing stratification boosting is also not clear to me. The loss function in Equation (4) involves the NLPL term, which should already separate the high-risk patients from the low-risk ones. I am not convinced that an additional term is necessary.

As shown in Table 1, the stratification boosting version does not outperform the DeepSurv version in cross-validation. However, it gives a much higher CI in the test set. The experimental results look suspicious to me and might need further investigation. The high CI in the small test set might be a coincidence and might not be reproducible with other test samples.

I am unable to understand Figure 2.

**Deanonymize Review:**

no

**Final Rating Justification:**

The authors have addressed my major concerns on clarity. I still doubt whether clustering helps for the method. But in general, I believe that it is reasonable to accept this paper.

**Justification Of The Preliminary Rating:**

I am unable to understand the motivation of the proposed method and some details about the formulation. The experimental results look suspicious because the test performance is much better than the cross-validation.

**Paper Type:**

methodological development

**Questions To Address In The Rebuttal:**

In Equation (4), how is the embedding $z_i$ related to the corresponding tile?

Is the cluster indicator updated during model training?

In Equation (4), how are the embeddings $z_i$ aggregated into the negative log partial likelihood (NLPL)?


**Special Issue:**

no

---

> ### Author Response · Authors · 2021-03-17
> **response to AnonReviewer3**
>
> Thank you for the thorough review of our paper. Based on your feedback, we clarified language and aligned mathematical notation to be consistent. Each question is addressed in detail below:
>
> 1. What is the motivation behind clustering?
>
> In our introduction, we say that subtyping is uniquely powerful for survival analysis, especially in the domain of cancer histopathology. We recognize that we should clarify “manual subtyping.” This is the traditional research practice for pathologists to qualitatively find recurring morphological features across a dataset to use as covariates in survival modelling. Tile clustering is done to replicate this within the machine learning domain using more automatic and quantitative methods. In previous work, we show that clustering without added supervised learning, as with EPIC, results in some clusters significantly contributing to survival prediction, by chance. With the added supervised function, we can guarantee that clusters, which represent morphology, do correlate to survival.
>
> We understand that some models learn directly from slide tiles. While simple aggregation methods such as mean or max might be good for cancer classification (even one tile is often enough to detect cancer), they fail in survival tasks.
>
> 2. Equation 4 is unclear (all equations are inconsistent)
>
> After review, we realize that this criticism was central to the lack of clarity in our paper. We apologize and have modified all notation to be consistent and further included notation in the model diagram (figure 2). You may find an updated PDF with corrected and consistent notation.
>
> Cox proportional hazard model predicts risk at time $t$: $\lambda(t) = \lambda_{o}e^{\beta_iv_i}$
> In the case of a neural network-based survival function, $\beta_i$ is substituted for model parameters, $\theta$, i.e. $\beta_iv_i \rightarrow f_\theta(S)$, where $S$ represents input slide images. (This sentence was added to the PDF)
> Equation 2 (NLPL loss) is rewritten using this terminology: $NLPL(f_\theta (S), d, e) = - \sum_{i:E_i=1} (f_\theta(S_i) - log\sum_{j\in\Re(T_i)}e^{f_\theta(S_j)})$
>
> In the body of the paper we clarify: “EPL models each WSI as $k$ groups of tiles with similar features, defined as parts, and backpropagates the loss against slide labels (time-to-event data) through the integrated encoding-aggregation graph, in which $k$ encoders ($\theta_e$) take in part representative tiles ($X$) and output part features ($z_i$) that are then concatenated and fed through a single fully connected aggregation layer ($\theta_a$).”
>
> Finally, we hope that these corrections make equation 4 more clear: $Loss = NLPL(f_\theta (S), d, e) + \lambda_c\sum_{i=1}^{N}||z_i-c_i||^2$ The second half of this equation (MSE) is also now shown to minize of distance between tile embedding $z_i$ and its associated part centroid $c_i$
>
> In the end, we hope these changes made our proposed method clear and the paper self-contained. Though, to save space, as mentioned in the paper, we suggest readers refer to EPL paper (Xie et al.) for even more details on mathematical induction and training iterations.
>
> 3. What is the motivation for stratification?
>
> DeepSurv, or more simply the Cox Proportional Hazard model, is defined to rank patients by risk of event. This ranking algorithm does not guarantee a strong separation between risk groups, which are necessary in the clinical setting.
> Stratification loss ensures a lower bound to the difference between the risk scores of two patients: $f_\theta(S_1) - f_\theta(S_2) > \epsilon, where \epsilon > 0$, whereas the deepsurv loss only guarantees the ranking of all slides $S$. In the clinical setting, patients are analyzed in relation to sub-populations rather than in relation to one continuous risk model. Making sure that the model can produce distinctly separate risk groups is important for building a model viable to the clinic.
>
> 5. Criticism that stratification boosting does not produce a significant improvement
>
> When using the concordance index as a metric, it is true that stratification boosting does not show a significant difference from DeepSurv. This is because CI is only measuring accuracy of ranking and not the distinction of risk groups. SB does not improve ordering but rather the distribution of ordering.
>
> The Kaplan-Meier curve helps visualize the power of stratification boosting. Specifically, at the 25% survival percentile (a standard metric), there is a significant survival difference measured in months. The KM plots on the test set indicate this difference as well.
>
> 5. Unable to understand figure 2
>
> We have provided mathematical notation within the figure to correlate with our clarified equations, as well as adjusted some of the second panel.
>
> 6. Do the clusters update?
>
> Refer to Fig. 2 caption “Global and local centroids are randomly initialized before training and updated between epochs”.
>
> PS We are in communication with our institution. Once we get approval, we will post full code.

---

> ### Author Response · Authors · 2021-03-19
> **view setting**
>
> We posted our rebuttal before the deadline (indicated on timestamp) but didn't realize that there was a "view" setting. We just changed the "view" to public (it was only viewable to program chairs and authors before). Apologies for that. Please consider reviewing our rebuttal.

---

### Author Response · Authors · 2021-03-17
**manuscript updated**

In addition to writing comments in response to great reviews of our work below, we have also modified the manuscript with the following changes:

- minor grammatical edits
- updated model diagram (figure 2) by adding mathematical notation and making some clarifying adjustments in the second panel
- adjusted mathematical notation to be consistent across all equations and writing

If accepted, we also plan to slightly extend discussion section to include comments we make in response to reviewers, as these discussion points are important. We also plan to resize figures to maximum resolution and extend their captions. For now, we are limited to 8 pages.

Thank you for all who contribute to review, criticism, and discussion

---

> ### Author Response · Authors · 2021-03-19
> **review view settings**
>
> We posted our rebuttal before the deadline but didn't realize that there was a "view" setting. We just changed the "view" to public (it was only viewable to program chairs and authors before). apologies for that

---

### Meta-Review · Area_Chair1 · 2021-03-27

**Recommendation:** Accept (Poster)

**Metareview:**

Three knowledgeable reviewers recommend weak accept after the rebuttal and discussion. One of the main concerns of the reviewers is the relatively small size of the dataset that is used in this study, however, all of them appreciated the presented method. I think that this paper will be a good contribution to MIDL 2021.

**Paper Type:**

both

---

### Decision · Program_Chairs · 2021-03-31

Accept